

# Assessing the skill of high-impact weather forecasts in southern South America: a study on Cut-off Lows

M. Belén Choquehuanca[1,2,3], Alejandro A. Godoy[4,5], Ramiro I. Saurral[1,2,3,6]

[1]Universidad de Buenos Aires. Facultad de Ciencias Exactas y Naturales, Departamento de Ciencias de la Atmósfera y los Océanos, Buenos Aires, Argentina
[2]CONICET-Universidad de Buenos Aires. Centro de Investigaciones del Mar y la Atmósfera, Buenos Aires, Argentina
[3]CNRS-IRD-CONICET. Instituto Franco-Argentino para el Estudio del Clima y sus Impactos (IRL 3351 IFAECI), Buenos Aires, Argentina
[4]Servicio Meteorológico Nacional (SMN), Buenos Aires, Argentina
[5]Facultad de Ciencias Astronómicas y Geofísicas (UNLP), La Plata, Argentina
[6]Barcelona Supercomputing Center (BSC), Barcelona, Spain

*Correspondence*: M. Belén Choquehuanca. (belen.choquehuanca@cima.fcen.uba.ar)

**Abstract.** Cut-off Lows (COL) are mid-tropospheric cyclonic systems that frequently form over southern South America, where they can cause high-impact precipitation events. However, their prediction remains a challenging task, even in state-of-the-art numerical weather prediction systems. In this study, we assess the skill of the Global Ensemble Forecasting System (GEFS) in predicting COL formation and evolution over the South American region where the highest frequency and intensity of such events is observed. The target season is austral autumn (March to May), in which the frequency of these events maximizes. Results show that GEFS is skillful in predicting the onset of COLs up to 3 days ahead, even though forecasts initialized up to 7 days ahead may provide hints of COL formation. We also find that as the lead time increases, GEFS is affected by a systematic bias in which the forecast tracks lay to the west of their observed positions. Analysis of two case studies provide useful information on the mechanisms explaining the documented errors. These are mainly related to the depth and the intensity of the cold core, which affect the thermodynamic instability patterns (thus shaping precipitation downstream) as well as the horizontal thermal advection which can act to reinforce or weaken the COLs. These results are expected to provide not only further insight into the physical processes at play in these forecasts, but also useful tools to be used in operational forecasting of these high-impact weather events over southern South America.

## 1 Introduction

Severe weather phenomena can significantly impact densely populated regions (e.g. Curtis et al., 2017; Newman and Noy, 2023; Sanuy et al., 2021). Over southern South America, these are frequently associated with heavy precipitation events triggered by low-pressure systems known as Cut-off Lows (COLs; Campetella and Possia 2007; Godoy et al., 2011a; Muños and Schultz, 2021). COLs are synoptic-scale weather systems that originate from elongated cold troughs in the middle





troposphere, which subsequently detach ('cut off') from the main westerly current (Palmén and Newton, 1969). This segregation from the main flow explains the isolated and erratic behavior of these systems, which suppose a significant challenge in operational weather forecasting, even for state-of-the-art numerical weather prediction (NWP) systems (Muofhe et al., 2020; Yánes-Morroni et al., 2018). Naturally, this can have an impact on the reliability of weather forecasts and early warnings which may be particularly relevant for southern South America considering the remarkable affectation from COLs (Godoy et al., 2011a).

Previous studies have focused on quantifying the explicit forecast errors associated with COLs in NWP systems. Gray et al. (2014) examined forecast ensembles from three operational forecast centers in the Northern Hemisphere and found that forecast errors were systematically larger in COL compared to no-COL events for the same prediction time. Similarly, Saucedo (2010) conducted an assessment of the prediction skill of the Global Forecast System (GFS) and Weather Research & Forecasting (WRF) models in southern South America for three COL events. His results indicated that forecast accuracy varies significantly depending on the individual COL cases and emphasized the need for an accurate representation of the COL center position during initialization to achieve better forecast results.

Other studies, such as those from Muofhe et al. (2020) and Binder et al. (2021), have linked errors in precipitation forecasts with inaccuracies in the location of the COL centers. In their evaluation of Météo-France forecasts, Binder et al. (2021) analyzed a single COL event and documented an eastward shift in both precipitation and COL position, primarily due to an initial underestimation of the COL intensity. Meanwhile, Muofhe et al. (2020) assessed the skill of the NWP model currently used operationally at the South African Weather Service to simulate five COL events. They observed variations in the predictive skill of COL-related precipitation across different development stages of the COLs, attributing these differences to inaccurate positioning of their centers. Moreover, studies by Bozkurt et al. (2016), Yánes-Morroni et al. (2018) and Portmann et al. (2020) have underscored the influence of the COL-induced circulation on extreme precipitation events, emphasizing the complexity and challenge of predicting these phenomena. In particular, Portmann et al. (2020) noted that uncertainties in the COL genesis position substantially affect the vertical thermal structure of a surface cyclone development as well as its subsequent evolution.

While previous studies have examined the skill of NWP systems in forecasting COLs, they usually cover a short period of time and do not address a compound evaluation of positional and intensity errors. For instance, the recent paper by Lupo et al. (2023) has quantified biases in COL forecasts globally, but for the operational version of the GFS model in a 7-year period running from 2015 to 2022. In this context, there is a necessity to deepen our comprehension of COL predictive skill, given the close linkage with heavy rainfall events. Our study tries to fill this gap, focusing on southern South America, a hotspot region for COL development (e.g., Reboita et al., 2010; Godoy, 2012 henceforth GD12; Pinheiro et al., 2017).

Our main goal is to assess the prediction skill of COLs in the National Centers for Environmental Prediction (NCEP)'s Global Ensemble Forecasting System (GEFS). This is achieved through quantifying forecast errors using an objective feature-tracking methodology which involves the identification and tracking of COLs along the forecast trajectories to produce a set of forecast versus observed COLs.



In this study, we specifically address three aspects of COLs: their onset time, their central position and their intensity. In
particular, we seek to respond the following questions:
1.  What is the temporal scale at which GEFS can reliably predict the initiation phase of COLs, and how precise are these

68          forecasts?

2.  After formation, can GEFS accurately predict the subsequent trajectories of the COLs?
3.  Can errors in COL forecasts impact those of precipitation further downstream?
It should be noted that this study can be considered as a first step towards a full characterization of the physical mechanisms
controlling the forecast skill of COLs and how the associated errors in state-of-the-art NWP systems are transferred into other
associated variables such as precipitation, atmospheric instability and winds. The rest of the paper is organized as follows: the
datasets and methodology are described in Section 2. The results on the forecast skill of the GEFS in both COL onset and their
evolution stages are included in Section 3, followed by a summary and the concluding remarks in Section 4.
**2 Data and methodology**
**2.1 The GEFS Reforecast dataset**
Daily averages from the GEFS Reforecast version 2 dataset (Hamill et al., 2013)  are used as a representative sample of the
GEFS model for the purpose of this study. This dataset consists of 11 ensemble members - one control run alongside 10
perturbed members - and covers a prediction horizon of 16 days after initialization. During the first week, data is saved at 3-
hourly intervals considering a horizontal resolution of T254 (roughly 40 km x 40 km at 40° latitude) and 42 vertical levels. In
the second week, the intervals increase to 6-hourly and the horizontal resolution decreases to T190 (around 54 km x 54 km at
40° latitude) with no changes in the number of vertical levels. The GEFS Reforecast dataset can be freely downloaded from
ftp://ftp.cdc.noaa.gov/Projects/Reforecast2, where the reforecasts have been saved at 1°x1° horizontal resolution from the
native resolution data using bilinear interpolation with wgrib2 software. It is worth noting that for all calculations within the
paper, we considered the ensemble mean as the basis for analysis and comparisons (i.e., no assessment is performed on
individual ensemble members). To validate the GEFS skill, we use the fifth version of the ECMWF Reanalysis Dataset (ERA5;
Hersbach et al. 2020) as a representation of the real-world conditions. The ERA5 data, with the original resolution of
approximately 0.25° x 0.25°, were coarsened to the same resolution of the reforecast to ease comparison.
Our analysis focused on the forecast verification of atmospheric variables at the 300 hPa level. This level was chosen because
it hosts both the largest frequencies and intensities of COLs within the Southern Hemisphere (e.g., Reboita et al., 2010; Pinheiro
et al., 2021). To detect COLs, we analyzed the geopotential height and the zonal wind component at 300 hPa as well as the
300/850 thickness. We also evaluated other variables of interest such as the geopotential height at 850 hPa and the total
accumulated precipitation to represent the lower-level circulation and related impacts of COLs.



## 2.2 Temporal domain and study area

The temporal domain of our study is based on the availability of reforecast data, ranging from 1985 to 2020. Specifically, we focus on the austral autumn season, covering the months of March, April, and May, which is the season with the highest frequency of COLs in South America (Reboita et. al., 2010; Pinheiro et. al., 2017; Muñoz et al., 2020). Regarding the spatial domain, we focused on the area of greatest occurrence of COLs, which encompasses the western side of southern South America (Reboita et al., 2010; Campetella and Possia, 2007; GD12). Specifically, we utilized the area situated between latitudes 37.6° and 29.9° S and longitudes 77.6° and 68.75° W, as illustrated in Fig. 1. This region has been extensively studied in the past by GD12, who found that the COLs in this area are particularly strong and can often cross the Andes Mountain range, leading to conditions prone to high-impact weather events over the continent further downstream (Godoy et al., 2011a).

## 2.3 COL identification and tracking algorithm

The methodology used to build the COL dataset from GEFS and ERA5 data aligns with the approach outlined by GD12 and underpinned by a detection algorithm grounded in the conceptual framework put forth by Nieto et al. (2005). The methodology looks for local minima in the 300 hPa geopotential height field by simply comparing the local height with neighboring grid points under certain restrictions of size (i.e. number of surrounding points) and intensity. When a minimum is detected, a second requirement is it being associated with a cold core, for which the 850/300 hPa layer thickness is considered as a proxy of the mean layer temperature. Finally, points that successfully passed criteria 1 and 2 must also be accompanied by easterly winds to their polar side to be labeled as a COL.

Once we identified the COLs, we tracked them using the nearest neighbor method in the GEFS and ERA5 datasets and determined their trajectories. Only those forecasted COLs that fill on specific matching criteria were retained for the subsequent statistical analysis. First, the trajectories were considered matched if at least one point (corresponding to one day) matched in time along the life cycle of the individual systems. Second, we state that the distance between the predicted and the observed first point of the trajectories should not exceed 800 kilometers. This distance choice corresponds to the typical diameter of COL systems, which is between 600 and 1200 kilometers (Kentarchos and Davies, 1998). In agreement with Froude et al. (2007), our spatial matching approach focuses primarily on the starting point of the predicted trajectories rather than the entire trajectory. This emphasis is because, although the trajectories may initially closely match the observed trajectories, they are likely to diverge as the forecast lead time increases.

## 2.4 Verification metrics

For the quantification of the model skill, we used a Lagrangian perspective to derive error statistics. This methodology has been previously employed to build position and intensity error statistics in previous investigations on tropical and extratropical cyclones such as in Froude et al. (2007) and Hamill et al. (2011). The validation metrics used in this study are sketched in Fig. 2 and are as follows:



●   Direct Positional Error (DPE): This metric is defined as the horizontal distance between the observed and forecast

127         positions at the same forecast time.

●   Cross-Track Error (CTE): This metric represents the component of DPE that is perpendicular to the observed track.

129         It provides information on the bias to the left or right of the observed track.

●   Along-Track Error (ATE): This metric represents the component of DPE that is along the observed track. It provides

131         information on the directional bias along the track, indicative of whether the forecasts predict a faster or slower motion

132         of the system compared to the reanalysis.

We adopted the convention that a positive (negative) value of CTE indicates a bias to the right (left) of the observed track,
while a positive (negative) value of ATE indicates that the forecast position is biased fast (slow). It is important to note that
CTE and ATE cannot be calculated for the first analyzed position of a COL since they depend on the existence of an observed
position the day before the valid time. For a more detailed explanation of these metrics, see Heming (2017).

## 3 Results

As a first step to determine the temporal horizon at which the GEFS model can forecast COLs, we analyze the central position
of the COLs and their intensity (given by the Laplacian of the geopotential height field). We show results only for the seven
days before the observed onset stage of COLs events since no COLs were detected beyond this period in the preliminary
analysis. It should be noted that hereafter "onset stage" or "onset" of the COL refers to the beginning of the segregation stage,
also known as stage 2 of the COL life cycle as defined by Nieto et al. (2005). We organized each forecast into eight groups
based on their initialization day, namely init 0, init 1, init 2, init 3, init 4, init 5, init 6, and init 7. Forecasts labeled as init 0
correspond to those initialized at the onset day of the COL, while forecasts labeled as init 1 to init 7 indicate forecasts initialized
one to seven days before the onset day of the COL, respectively.

### 3.1 Predictive skill of COL onset time in GEFS

Figure 3 shows the percentage of detected COLs as a function of their initialization day, i.e. how many days in advance could
these systems be forecasted in the GEFS dataset. During initializations closest to the onset days (init 0 to init 2), over 94% of
the total events (32 out of 34 COLs) were accurately predicted by the GEFS. However, this accuracy decreases significantly
from init 3 onwards: 71% at init 3, 56% at init 4 and down to only 9% at init 7. It is interesting to highlight, still, that the
reforecasts were able to correctly predict most COLs on the same date they were observed, even when the initializations were
farthest from the onset days (i.e. init 4 and init 5), indicating the accuracy of GEFS for predicting the timing of the events.
Figure 4 illustrates the quartile distribution of the DPE and intensity error in the GEFS model for the onset day of the COLs
where each boxplot represents a different initialization day. The boxes represent the interquartile range (IQR), which comprises
50% of the error distribution, with the median value indicated by a bold black line. Initially, a gradual increase in the median
of DPE can be observed as the number of days before the onset of COL increases (Fig. 4a). The DPE increase varies from 140





kilometers at the first initialization (init 0) to about 300 kilometers at init 3. At the same time, the IQR expands from 300
kilometers at init 1 to 900 kilometers at init 3, indicating a widening spread of DPE with increasing forecast time. In contrast,
the median of the intensity error exhibits a negative trend: it decreases from -2.5 gpm/m$^2$ at init 1 to -8 gpm/m$^2$ at init 3, with
an IQR that varies significantly with the day of initialization. For subsequent initializations (init 5 to init 7), we observe a
continuous increase in DPE from 400 kilometers to approximately 600 kilometers, alongside a consistent negative trend in
intensity errors, with values around -13.0 gpm/m$^2$. However, it is important to note that these results are based on a smaller
sample size than previous initializations and caution should be exercised when generalizing these results.
Figure 5 shows eight polar scatter plots illustrating the errors in the position of the predicted COLs in comparison to the
reanalysis, with each plot corresponding to a particular initialization day. During the early initializations, the GEFS exhibits
errors contained within a radius of 3° (approximately 300 km) around the observed positions and shows no discernible
directional deviation. This indicates that the position errors are randomly distributed and show no systematic bias, which is
particularly clear up to init 2. Conversely, initializations from init 3 to init 5 show a larger spread, with more points deviating
significantly from the observed cyclone positions. While we detected a southward deviation, the zonal (i.e. east-west) behavior
was less uniform, as init 3 showed a southern bias, init 4, a southwestern bias, and init 5, a slight southwestern deviation. This
indicates overall a slight deviation towards the south (on average between 1° and 3°), even if there is no clear longitudinal
direction. Forecasts initialized with a larger lead time showed a larger spread, partly due to a smaller number of predicted
COLs, but also revealing a predominant southwesterly bias of the model.

## 3.2 Predictive skill of COL intensity and tracks in GEFS

In this section, we investigate whether there is any bias in predicting cyclone intensity, propagation speed, and trajectory. We
focused on the forecasts initialized up to 3 days before the segregation date since the number of detected cases is significantly
lower for forecasts initialized beyond that point (i.e. init 4 to init 7), as explained in Figure 3. Also, considering that most COLs
have a duration of 4-5 days or less (not shown), we restricted our analysis to forecast lead times within 3 days of the detection
of the COLs in the ERA5 reanalysis.
Figure 6 shows the quartile distribution of DPE between the GEFS and ERA5 trajectories for init 0 to init 3. Each initialization
shows similar sensitivity: in the case of init 1 and init 2 (Fig. 6b,c), errors increase from 166 to over 320 kilometers within two
or three days after COL detection in the ERA reanalysis. The situation is similar for init 0 (Fig. 6a), where the error increases
from 144 to over 275 kilometers in the same period. Not surprisingly, init 3 (Fig. 6d) has the largest mean error, with a linear
increase from 290 to 550 kilometers. As regards IQR, it shows a linear increase, indicating that the dispersion of the position
errors increases along the cyclone forecast period.
Conversely, a negative trend is observed in the intensity difference between the matched GEFS trajectories and the
corresponding ERA5 reanalysis trajectories (Figure 7). The error for init 0 and init 1 (Fig. 7a,b) initially increases from -2.0 to
over -4.3 gpm/m$^2$ within two to three days after COL detection in the ERA reanalysis. For init 2 and init 3 (Fig. 7c,d), however,
a further increase in the error can be observed. While init 2 shows an increase from -4.9 to -11.68, init 3 shows an even more



pronounced initial error of -8.14, which subsequently increases to -9.0. Regarding the dispersion of the error, it is noteworthy
that init 1 and init 2 (Fig. 7b,c) show a slightly positive trend, indicating an increase in the uncertainty of the predicted system
intensity. In contrast, the last initialization (Fig. 7d) shows a significantly larger dispersion and a more variable behavior during
the analyzed period. Despite the observed variability, however, a trend towards greater dispersion is discernible.
Given that the DPE may stem from biases in either the translation speed of the COL (ATE) or from its direction of motion
(CTE), as shown graphically in Fig. 2, we disaggregate their relative contributions in Figs. 8 and 9, respectively. In general,
the ATE distribution exhibits a negative bias towards the later stages of the forecast tracks, except for init 2 (Fig. 8c) which
shows slightly positive values. Both init 1 and init 3 (Fig.8b,d) exhibit negative biases with median distances of around 200
and 300 kilometers, respectively. This negative bias in ATE may indicate that GEFS tends to underestimate the translational
speeds of COL towards the latter stages of the forecast lead times. Regarding the CTE distribution (Fig. 9), no clear bias is
observed, however, there are some noticeable trends in different initializations. In particular, init 2 (Fig. 9c) shows negative
values at around 100 kilometers. On the other hand, init. 3 (Fig. 9d) displays predominantly positive values, representing a
poleward bias according to its definition.

### 3.3 Case studies

In this subsection, we focus on two COLs that exhibited very different levels of prediction performance during their onset
stage (Fig. 4a). The first case study, from March-April 2013, is characterized by small DPE values, below the first quartile in
Fig. 4a, indicative of a forecast with high accuracy in the GEFS dataset. In contrast, the second case study, from March 2019,
was associated with remarkably larger DPE values, with errors ranging between the median and the third quartile. This
represents a scenario in which the prediction has a suboptimal performance. It should be noted that the selection of the case
studies was based also on the impact the model errors had on the associated precipitation downstream. Before exploring the
associated errors in the GEFS dataset, we provide a brief description of the synoptic environment around each COL during its
segregation stage.

### 3.3.1 Case study 1: COL development on March 31st, 2013

On March 31st, 2013, a COL formed to the west of the Andes Mountains at 36°S and 75.5°W. Its lifespan lasted for six days,
covering a distance of over 2,000 kilometers into the Atlantic Ocean (not shown). This event was associated with severe
weather conditions which resulted in unprecedented flash floods in the region, leading to loss of lives, significant infrastructural
damage and economic losses of USD 1.3 billion (Pink, 2018).
During the segregation phase of the COL, the main atmospheric features included an amplified ridge downstream of the system,
the presence of two jet streaks - one to the north and one to the south of the COL - and a well-defined cold-core in the middle
levels, and a cyclonic system off the central coast of Chile at lower levels. The circulation of the cyclonic system (Fig 10c)
fostered cold air advection underneath the COL center which helped to sustain and intensify the COL itself (not shown). During



its early development stages, this COL led to record-breaking rainfall of over 25 mm per day with peaks in excess of 50 mm
in certain areas over south-central South America ( Fig. 10b).
Forecast-wise, it is found that the location of the COL formation was accurately predicted 1 and 3 days ahead (init 1 and init
3; second and third rows in Fig. 10), but both initializations underestimated its intensity by approximately 6 gpm/m$^2$ and 11
gpm/m$^2$ in init 1 and init 3, respectively. The GEFS accurately predicted the strength and extent of upper high winds associated
with the COL (Fig. 10d,g). However, it underestimated the strength of the cold core in middle levels and misplaced the location
of the cyclonic circulation at lower levels, which shifted to the north of the observation site (Fig. 10f,i). This suggests that the
vertical coupling with the COL was affected, potentially impacting the intensity of the system. Regarding precipitation
forecasts, in both init 1 and init 3, the regions with significant rainfall were located southeast of their actual position and
amounts were overall underestimated, particularly in init 3. On the other hand, init 5 exhibited even less skill, with intensity
and location errors of around -14 gpm/m$^2$ and 200 kilometers northwest of its observed position, respectively. The GEFS also
encountered difficulties in predicting the jet split structure, inadequately represented the low-level circulation, and failed to
capture the cold core at mid-levels, which naturally had an impact on the predicted precipitation amounts as well (Fig. 10k).
Rainfall forecasts located the highest precipitation in the northeast of the country, outside the area affected by the COL system.
This suggests that GEFS may not perform well in producing precipitation associated with COLs.
**3.3.2 Case study 2: COL development on March 9th, 2019**
On March 9th, 2019, another COL formed off the coast of Chile, at 33°S and 74°W (first row of Fig. 11). This system was
weaker than the previous COL and lasted for four days. It caused some weak precipitation in south-central South America, but
the amounts were lower than those associated with the first COL.
The synoptic environment during the segregation stage of this COL in the ERA5 reanalysis (first row of Fig. 11) included an
upper-level ridge with a NW-SE axis to the southwest of the COL, a split jet structure, a strong low-level cyclone positioned
just beneath the COL center off the coast of Chile, and a small cold core at middle levels. Although this COL had a smaller
structure than the first COL, the cyclonic system extended into the lower levels, as evidenced by the accompanying low-level
cyclone identified in Fig. 11c. In the precipitation field, two distinct maxima were observed: one situated northeast of the
domain of analysis, probably linked to a decaying frontal zone over that area (not shown), and another one over western
Argentina related to the ascent zone at the east of the COL. The subsequent validation of the GEFS forecast focuses only on
this second feature as it was the one directly associated with (or triggered by) the COL.
The GEFS forecasts for March 9th, 2019 initialized 1, 3 and 5 days ahead are shown in Fig. 11 (second to fourth rows). In init
1 (Fig. 11, second row) the forecasted COL was approximately 15 gpm/m$^2$ shallower and located around 210 kilometers to the
southeast compared to ERA5. Regarding the circulation at upper levels, GEFS predicted well the strength and extent of high
winds associated with the COL. However, the circulation at low and middle levels was less accurate. GEFS predicted the
850hPa cyclone to be located further north than expected, and at middle levels, it failed to represent the cold core. For init 3
(Fig. 11, third row), the forecasted COL was approximately 17 gpm/m$^2$ shallower and 430 kilometers southeast of its actual



intensity and location. In this case, while GEFS predicted well the strength of the winds associated with COL, their position
was predicted wrong,  eastward compared to its actual position. At low and mid-levels, the forecast was also inaccurate; the
850 hPa cyclone was weaker and displaced more northward than observed, and the strength of the cold trough at middle levels
was underestimated and displaced towards the east. Regarding rainfall amounts, both initializations underestimated the rainfall
within the ascent zone of the COL and predicted to be northeast of their observed position, over the central and northeastern
parts of the country (Figure 11e, h). As for the last initialization (Fig. 11, fourth row), the model failed to predict the COL.
GEFS displaced the upper circulation towards the southeast, including the jets and associated upper ridge. At low levels, GEFS
also failed to predict the cyclone off the coast of Chile. Meanwhile, the thickness field showed a small, less intense cold trough,
resulting in a lack of rainfall amounts over the zone influenced by the COL, as shown in Fig. 11k.
Based on these results, a wrongly positioned and less intense COL can lead to a poor forecast of the cold core, subsequently
affecting dynamical processes such as horizontal temperature advection, thermodynamic instability, vorticity advection and
associated ascent which are ingredients for precipitation production downstream. Such errors may be related to the inadequate
representation of diabatic effects or interaction with the Andes Cordillera (Garreaud and Fuenzalida 2007). However, it is
beyond the scope of this study to draw conclusions regarding how GEFS simulates the processes associated with COLs.
**4 Discussion and Conclusions**
This study explored the prediction skill of cut-off lows (COLs) in the NCEP Global Ensemble Forecasting System (GEFS)
with a focus on the region with the highest frequency of COL occurrence in South America during austral autumn (March to
May). The analysis made use of a verification framework centered on the individual systems.  These were identified and
tracked using a feature-based approach applied to the 300 hPa level geopotential height as the primary variable.
The main conclusions can be built on the questions posed at the Introduction of the study:
What is the temporal scale at which GEFS can reliably predict the initiation phase of COLs, and how precise are these
forecasts?
The GEFS model is highly accurate in predicting the start of the segregation stage of COLs up to three days in advance, but
this accuracy drops significantly as the lead time increases beyond four days. The percentage of COLs detected by the model
decreases to 56% and 29% for predictions initialized four and seven days ahead of the segregation, respectively. Our analysis
also revealed that  COL centers diverge by an approximate distance of 200 km relative to the observations up to three days in
advance. However, this error increases to 600 kilometers for forecasts more than four days ahead. Also, it has been shown that
forecasts initialized up to two days in advance have no directional deviations while forecasts initialized at least three days
ahead of COL formation have a predominant southerly bias. At the same time, the intensity errors show a consistent increase
in magnitude, with values ranging from -2.5 gpm/m$^2$ in init 1 to approximately -13.0 gpm/m$^2$ at higher lead times.
After formation, can GEFS accurately predict the subsequent trajectories of the COLs?



From our results, we can conclude that the GEFS model has variable skill when forecasting the trajectories of COLs. Overall,
errors increase from 200 to 400 kilometers in position in forecasts of one to two days of lead time. Within this time period,
trajectories tend to be slower in comparison to the observed behavior. Even though this pattern of errors is also found for
longer lead times, errors in predictions three days ahead increase substantially, and skill beyond four days is dramatically
reduced. We can conclude that the trajectories of COLs can be relatively well predicted with lead times up to three days, and
forecasts initialized beyond that threshold are significantly degraded and depict a poor representation of the actual paths.
Intensity-wise, we found that GEFS forecasts are characterized by an increase in the magnitude of underestimation of COL
intensity as the lead time increases.
Can errors in COL forecasts impact those of precipitation further downstream?
Even though in this study we have provided only partial evidence on this point from the analysis of two case studies, we can
conclude that in these events the predictive skill of COLs (including their formation location, intensity and trajectory) had a
significant impact on the precipitation forecasts downstream. In particular, the errors in the location and depth of the COLs
were linked to the mechanism sustaining these systems, ~~among which the thermodynamic instability played a role~~. For
example, underestimating the strength of the cold core of COLs can significantly alter thermodynamic instability patterns,
affecting vertical motion and precipitation formation downstream. Moreover, incorrectly forecasting the position of a low-
level cyclonic system in association with COLs can significantly impact the vertical coupling of COLs, potentially influencing
their intensity. This aligns well with Pinheiro et al. (2021), who suggested a possible relation between the intensity of COLs
in South America and their vertical depth. These deficiencies, transferred into the higher levels, are able to shape the intensity
of the system and, via this alteration, some of the mechanisms responsible for precipitation formation. For instance, a weaker
(stronger) COL will foster more (less) vorticity advection, resulting in favored (unfavored) ascent downstream. Therefore,
predicted precipitation amounts will naturally be modulated by these errors (e.g. Saucedo, 2010).
Results from this study can be compared with similar recent studies. For instance, Lupo et al. (2023) have concluded that the
operational GFS model has a systematic bias to move Southern Hemisphere troughs and COLs too quickly downstream, even
though in our study region the identified bias is towards the west (i.e. slower than observed). It should be noted, however, that
the GEFS and the operational GFS share some common components but are different models, particularly regarding the
horizontal resolution. As such, results from both studies are not directly comparable.
Regarding the case studies, previous authors analyzing the synoptic evolution and predictive skill of COLs in other regions of
the world, such as Portman et al. (2022) and Moufhe et al. (2020), have concluded that a proper representation of the vertical
structure of the COL is crucial for a proper representation and prediction of these systems. Pinheiro et al. (2021) also argue
that the intensity of the COLs affect the entire structure of these systems, and that errors in their intensity/position can easily
affect their associated precipitation fields.
It should be stressed once again that this study is proposed as a first step towards a full characterization of the physical processes
responsible for COL formation, evolution and predictive skill in NWP systems. Several open questions remain, which will be
addressed in future studies. Among them, it is unclear why the predicted trajectories are systematically slower than the



observations. A negative correspondence between COL intensity and location was also observed in the GEFS dataset,
suggesting that the most intense COLs seem to be associated with lower positional errors. However, the underlying mechanism
sustaining such a relationship (if any) is not clear.
As a final note, future studies will dive into the relative contributions of COL intensity, location and speed on the resulting
forecasted precipitation fields, as a deeper understanding of the interplay between these might bring useful information for
operational weather predictions of high-impact events over southern South America.
*Data availability*. All data is available from the authors upon request
*Author contributions.* BC prepared all analyses and the manuscript. AAG provided scientific advice throughout the whole
project and assisted in setting up the tracking algorithm. RIS provided editing assistance, technical support and valuable
suggestions for improving the manuscript.
*Competing interests*. The authors declare that they have no conflict of interest.
*Acknowledgements.* The authors would like to acknowledge the European Centre for Medium-Range Weather Forecasts
(ECMWF) as well as NOAA's Earth System Research Laboratory (ESRL) for making available the ERA-Interim and GEFS
datasets used in this study. Grant PICT2020-SerieA-03172 from the Agencia Nacional de Promoción de la Investigación, el
Desarrollo Tecnológico y la Innovación (Scientific and Technological Ministry of Argentina) to the University of Buenos Aires
partially funded this research.

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



**Figures**

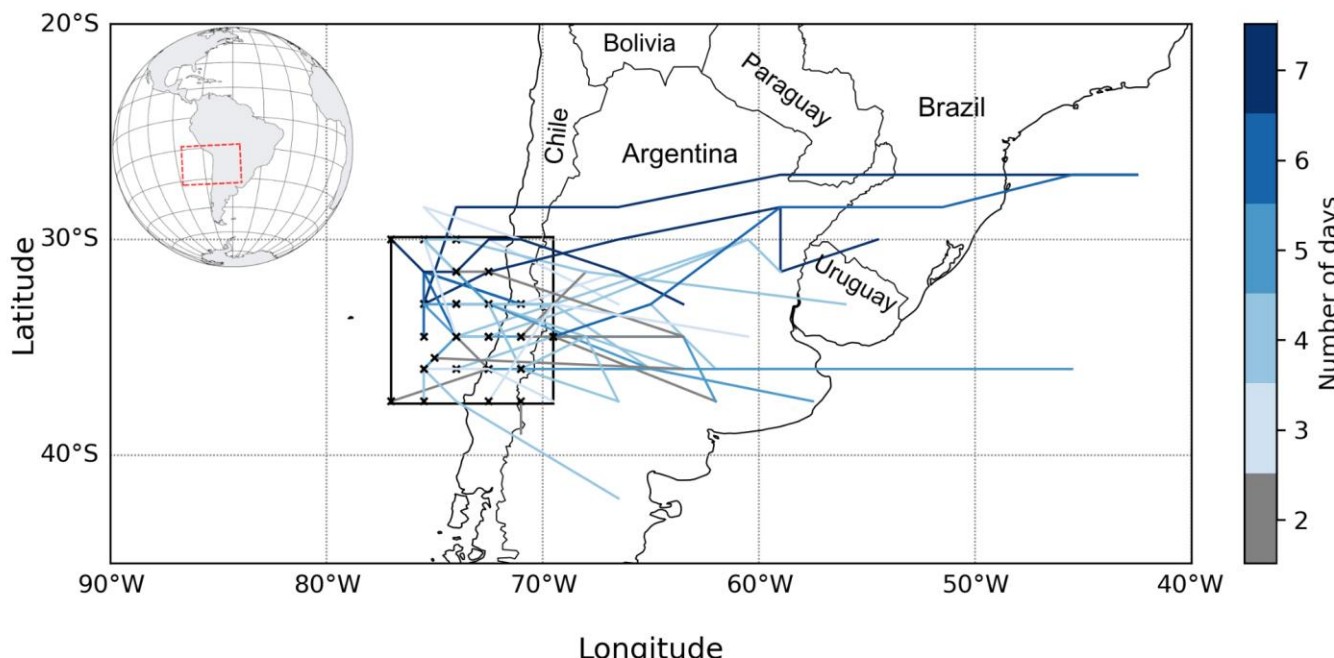

Figure 1: Spatial distribution of COLs in the region of highest COL frequency in southern South America from 1985 to 2020. Black crosses represent the start of trajectories of COLs detected in the study area (77.6°-68.75°W and 37.6°-29.9°S, solid black box) and lines represent their trajectories where colors represent the duration of each COL.



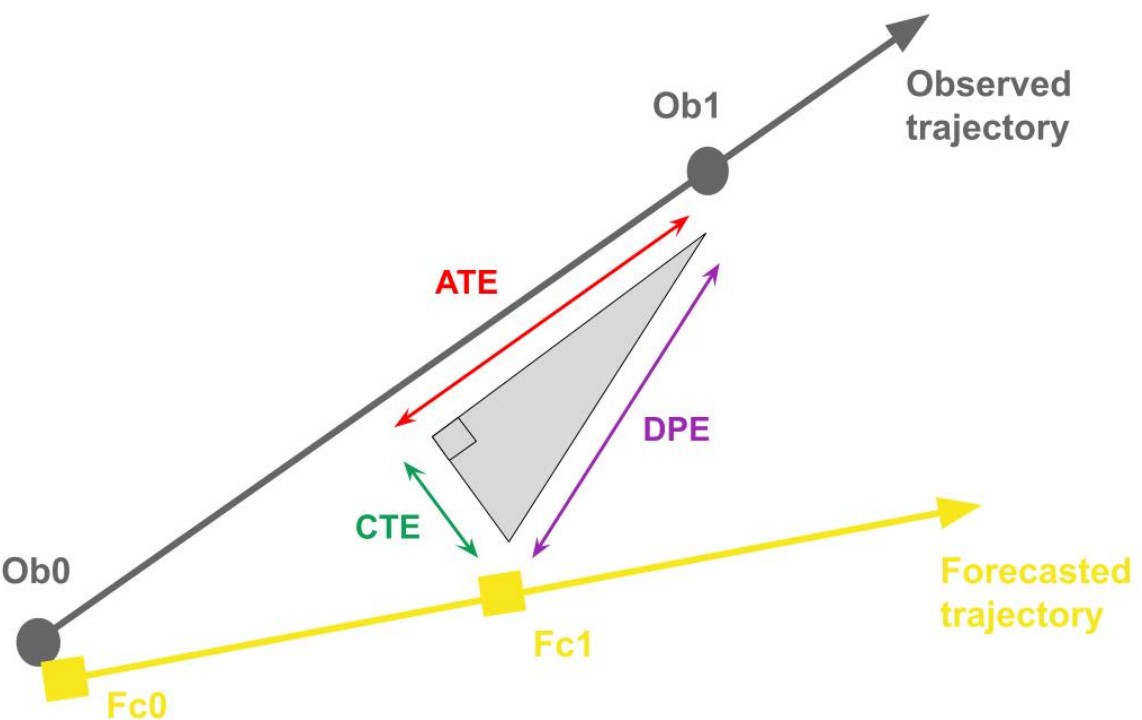

442

**Figure 2: Measures of cyclone track forecast error: Direct Positional Error (DPE; violet arrow), Cross-Track Error (CTE; green arrow) and Along-Track Error (ATE; red arrow). Obs0 and Obs1 are observed positions at times 0 and 1, while Fc0 and Fc1 are their respective forecasted positions. The gray circles (yellow squares) represent the observations (the forecasts).**



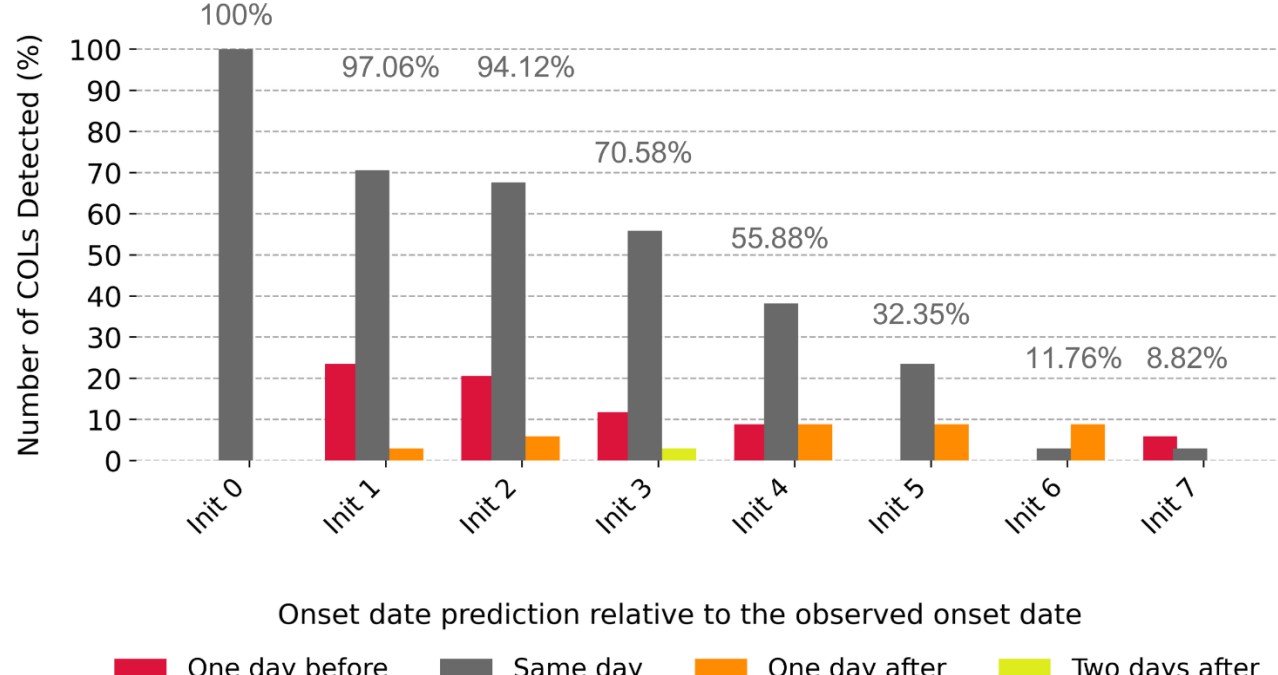

**Figure 3: Percentage of forecasted COL initiations as a function of initializations, from init 0 (forecast initialized in the onset day) to init 7 (forecast initialized seven days before the onset of the COL). The red, gray, orange and yellow bars indicate the forecasted date of the onset day of COL relative to the observed date of onset day, from one day ahead of formation to two days after, respectively.**



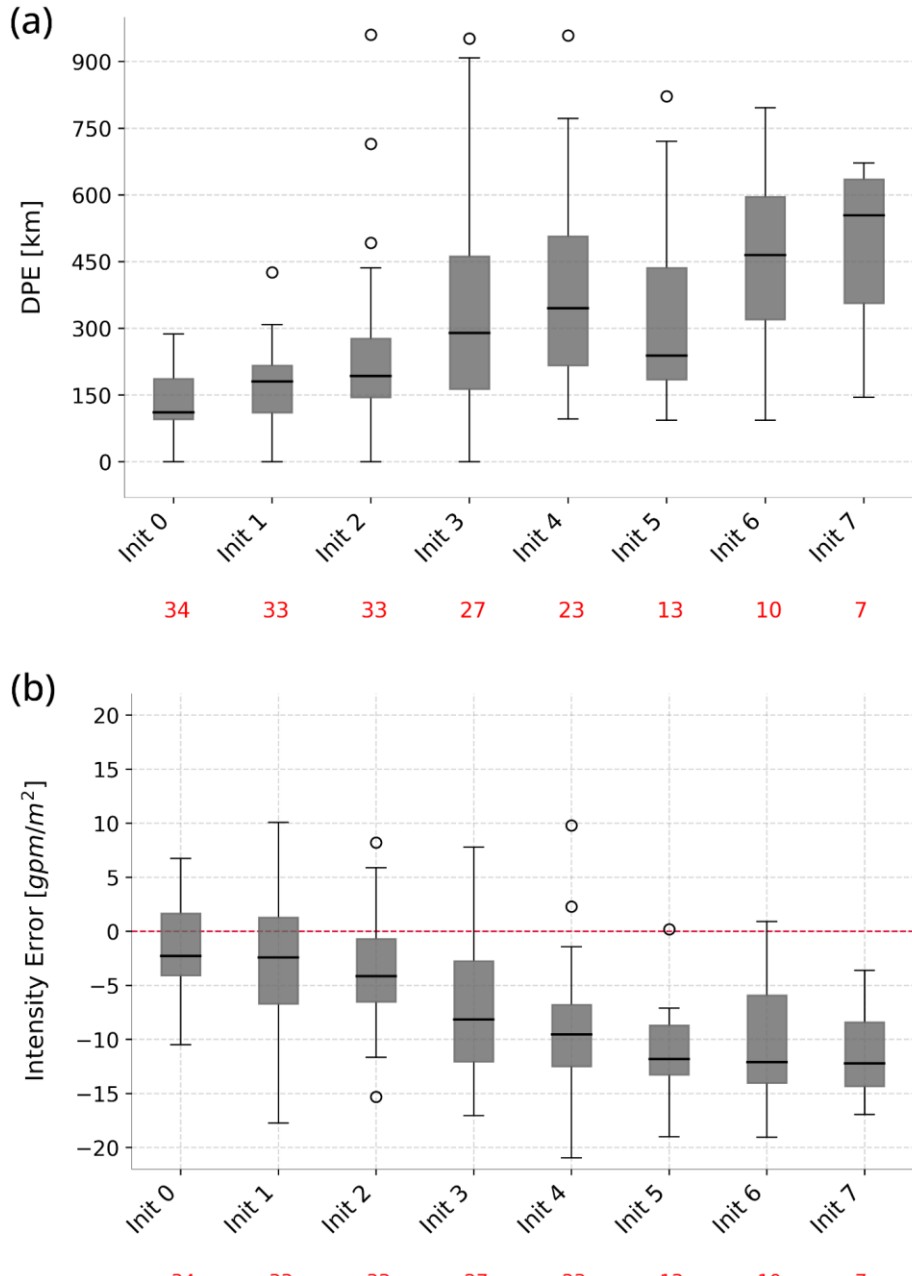

451

**Figure 4: Variation in a) onset position (DPE) and b) the intensity error as a function of initializations. The whiskers at the top (bottom) of the boxes represent the error's 75th (25th) quantile. The black thick horizontal lines inside the boxes represent the median (the 50th quantile) and the points outside the whiskers are considered outliers. The red numbers at the bottom indicate the number of systems identified under each initialization.**



**Figure 5: Scatter diagrams of COL initial position deviation decomposed in longitudinal and latitudinal errors (in degrees), where the central axis is the initial position observed. Each plot represents a different initialization: ranging from a) Init 0 (forecast initialized in the onset day) to h) Init7 (7 days in advance). The gray/black dots indicate the location of the predicted COLs as a function of the initialization day (see the color bar for reference on the number of predicted systems per day). The red dots show the mean location after averaging all the COLs predicted in each initialization day.**





**Figure 6: Boxplots of DPE tracks along the life cycle of the COLs, where each plot represents initializations at a) Init 0, b) Init 1, c) Init 2, and d) Init 3. The red numbers at the top indicate the number of systems identified for each forecast lead time.**







465

**Figure 7: As in  Figure 7 but for the intensity error.**







**Figure 8: As in Figure 7 but for the along-track error ATE.**







469

**Figure 9: As in Figure 7 but for the cross-track error CTE.**





**Figure 10: Segregation stage of the COL formed on March 31st, 2013. (Top) ERA5 and (rows 2 to 4) GEFS predictions of (first column) geopotential height (Z) and wind (U) at 300 hPa, (second column) geopotential height (Z) at 300 hPa and accumulated precipitation (Accum. prep.) over 24 hours, and (right column) geopotential height (Z) at 850 hPa alongside the 850/3000 hPa layer thickness (DZ) GEFS predictions correspond to init 1 (second row), init 3 (third row) and init 5 (fourth row), initialized on March 30th, March 28th and March 26th, 2013, respectively.**









**Figure 11: As in Figure 11 but for the COL formed on March 9th, 2019. In this case, the GEFS predictions corresponding to init 1 (second row), init 3 (third row) and init 5 (fourth row) were initialized on March 8th, March 6th and March 4th, 2019, respectively.**