# Peer review of "Assessing the skill of high-impact weather forecasts in southern South"

_EGUsphere, 2024_

## Referee Comment (RC2)

**Review of "Assessing the skill of high-impact weather forecast in southern South America: a study on Cut-off Lows" by Belén Choquehuanca et al. (2024)**

This article analyses the predictability of cut-off lows during Autumn that frequent the south-west coast of South America. The authors analyse the forecast lead-time of cut-off low formation identified objectively in GEFS forecast data. Understanding the predictability of cut-off lows is important to understand, particularly in this part of the world, where they have high impact, as the authors suggest. There are however some gaps in how various analyses and processes are discussed in this work that need to be improved before publication.

Some general, specific and technical comments are below.

**General comments**

1. Methods non-specific (Section 2.3)
   The methodologies used within this study need to be clarified. I can understand that they are based on methodologies used and referenced in previous work, however at least some minor detail needs to be added to contextualise the results for the reader. This is especially true in Section 2.3. For example, "certain restrictions" - what are they?; "winds on the polar side of the labelled COL" - how far poleward?. These details are important for the reader to understand in order to understand the result presented.

2. Verification metrics used
   I have concerns that some of the verification metrics used may not be reflective of the processes they are trying to verify. This is particularly true of the CTE and ATE metrics. I am not sure that the ATE will consistently show an error in the speed of the track and CTE an error in the positional error of the track. An example situation of this is described in the specific comments below. Would showing speed and bearing not be more appropriate?

3. Sample size
   The sample of COLs used in this work seems incredibly small given the number of COLs that this region receives - this represents less than 1 COL per year. I wonder therefore if the region or criteria is too strong in this study. Relaxing it and getting more events would help significantly in making sure the results are more robust. I was also unclear exactly how COLs in the study region are selected - is it only COLs that begin (as Stage 2) in the region that are selected? Or are COLs that move through the region as Stage 2 also counted?

4. Some conclusions reached seem to lack evidence
   The authors make several conclusions related to the precipitation patterns, depth or vertical coupling and thermodynamics of the COLs related to errors in the forecast of these systems. Many of these are expressed as important results in both the conclusion and the abstract, however, in my view, these largely have very little evidence to back them up. These are largely based on two individual case studies of COL simulations by analysis of the ensemble mean. Can holistic conclusions be drawn for just these two cases? Statements such as "predictive skill of COLs had a significant impact on the precipitation forecasts downstream" or "underestimating the strength of the cold core of COLs can significantly alter the vertical coupling of COLs" need to be robustly backed up by the results. One way to make this more robust

would be analyse each of the members and leverage the ensemble spread to understand the processes associated with small and large positional errors.

**Specific comments**

- L70: The precipitation patterns studied seem very local and not "downstream". In my view therefore, this posed question is not answered as stated.
- L88: Previous authors (e.g. Reboita et al. (2010)) have found large differences in frequencies of COLs for different reanalyses, with ERA reanalyses producing more COLs than NCEP reanalyses. Since the authors use GEFS, would the use of NCEP reanalysis not be a fairer comparison?
- L90-93: Does the orography affect your choice of using 850hPa in your thickness criteria? Presumably are large proportion of the Andes is generally above 850hPa.
- L94: I assume the "accumulated precipitation" is ERA5 data. General I have found that these totals are generally error prone. Is ERA5 precipitation reliable?
- L108: What are these restrictions?
- L109: I assume you look at a minimum in thickness? Provide details on this procedure
- L111: How far poleward?
- L113: What are these specific criteria?
- L117-120: Not sure I understand this point. Consider clarifying.
- L128-132: How robust is the use of ATE to represent speed and CTE to represent spatial error? I can envisage situations in which the along track error vastly under-estimates the speed of propagation (see my poor recreation of your Fig 2. below). In this case, the speed error is presumably under-estimated if one looks at the ATE, however in reality the speed is actually faster than in reality. Would using speed/distance travelled (to determine speed errors) and bearing (to represent bias in left/right not be better?

[Figure]

- L139: How is intensity defined in this study?
- L139-141: "7 days before" - Do you refer to forecast lead-times here? Clarify this statement.
- L149: 34 COLs - seems a small number (~1 COL per season). Would increasing your domain size give you more COLs and increase the robustness of your results? This number is too small to get a robust result in my view
- Figs 6-9: I strongly suggest that these figures are combined into a single plot to help the reader compare these different metrics. Either stack each metric on an init axis,or stack inits on a metric axis.
- L178-179: Do all your tracks last at least 3 days in order to study the track evolution after 3 days? Most climatologies (some of which are cited here) shows the majority of COLs last for only 1-2 days. Please clarify.
- Section 3.2: Forecast COLs are generally weaker than observed. How does this impact their predictability? Do they generally terminate more quickly than observed as they are weaker?
- L192: "except for init 2" - init 0 also fairly symmetric
- L217-220 and 240-243: This synoptic description is unnecessary. This circulation is there by definition of your COL identification technique.
- L226-228: It is unclear from your analysis how the vertical coupling was affected by the COLs incorrect forecast position. Be specific and add detail to this analysis
- Figures 10-11: Both Sections 3.3.1 and 3.3.2 make reference to low-level circulation patterns, however these are not shown.
- L231-233: I disagree from Figure 10, that one could say that GEFS had "difficulties" in predicting the split jet and "failed" to capture the cold-core. Both these features are present in Figure 10j,k,l
- L235: Do you feel that this is a robust statement? In general GEFS seems to have done a decent job. Your longest lead-time produced a weaker COL in the mean which affected the forecast of higher precipitation. However I feel that the statement that GEFS (which is relatively course resolution here) may not perform well for COL precipitation forecasts is very harsh given the data provided and the severely limited cases looked at.
- L244-246: Is this frontal zone not linked at all to the COL? Ie. Is this front not connected to the surface low (if there is one) associated with the COL or is it linked to a completely separate mid-latitude cyclone? Be specific.
- L251: "the circulation at low and mid-levels" - how do we know this? This implies that the low and mid-level circulation are completely independent of the upper-levels, which they are are not of course. Clarify and add detail to explain your analysis.
- L257-259: Similarly to case 1, I think the rainfall patterns presented look pretty good, especially considering that the model is 1x1 degree. What is the effect of the resolution on your analysis?
- L262: The final row of Fig 11 appears not to be cut-off, but called a COL here. Would this have been detected as a COL in your analysis?

**Technical comments**
- L32: "suppose" -> "pose"?
- L82-83: is data from the second week used here? It seems that only lead-times up to 7 days are investigated. If not, remove this detail.
- L137: "position is biased fast (slow)" - consider rewording

- L168: "Conversely,..." - this sentence is not converse to the previous statements. Both indicate large spread, the one is simply larger than the other.
- L172: "direction" -> "bias"?
- Figure 10-11: Maybe "Obs" is not a representative label, but rather "Reanalysis".
- L176 and elsewhere: You use the terminology "segregation", "onset" and "start" throughout the article. It remains unclear whether these are different or mean the same thing. If there are the same thing, stick to one terminology. If not, define them explicitly and clarify.
- L182: "ERA" -> "ERA5"?
- L187-202: "increases" - do you mean "decreases"? The value gets less. The magnitude of the error increases. Be specific.
- L221: "record-breaking rainfall": 25-50mm seems low for a rainfall record. If this is the record, then add a reference to backup this statement.
- L243: "northeast of the country" - which country?
- L238: "the previous COL" -> "Case 1"
- L243-244: "Figure 11c" - this is thickness and geopotential height at 300hPa and not low-level circulation as specified.
- L282: "have a prominent" -> "tend to have a" - the bias is not that prominent is it?

---

## Author Comment (AC1)

**Authors' response to reviewers' comments**

Manuscript Number: egusphere-2024-1063

Assessing the skill of high-impact weather forecasts in southern South America: a study on Cut-off Lows

The authors sincerely thank both reviewers for their detailed and constructive feedback, which significantly improved the manuscript. We have carefully considered all the comments and made substantial revisions to address the issues raised. These revisions have strengthened the analysis, particularly in relation to the predictability of Cut-Off Lows (COLs) and their impacts on Southern America during austral autumn. We believe the revised manuscript is now significantly clearer and more impactful.

In this document, the responses to the reviewers' comments are highlighted in red font, and the resulting corrected text is highlighted in green

**Reviewer 1**

**Summary and recommendation**

Cut-off lows (COLs) are important circulation systems that can often bring extreme precipitation. Using the Global Ensemble Forecasting System, the authors systematically assess the prediction skill of cut-off lows over Southern America during austral autumn. The showed that the COLs can be successfully predicted 3 days ahead. The model ensemble mean tends to underestimate the intensity and has track bias to the west. This provides a comprehensive assessment of the COL skills by current ensemble forecast systems. The results are clearly stated and I suggest minor revisions.

**Minor comments**

1. **Minor comment 1:** I think the current manuscript is missing a discussion of the physical processes responsible for the COLs in Southern America. This is important because this may help readers to understand why the models tend to underestimate the COLs and have track bias. For example, does the bias result from the weak eddy-mean flow interaction in the evolution of COLs (e.g., Pinheiro et al., 2022; Nie et al, 2022, 2023)?

   References to minor comment 1:

   - Nie, Yu, Jie Wu, Jinqing Zuo, Hong-Li Ren, Adam Scaife, Nick Dunstone, and Steven Hardiman, 2023: Subseasonal Prediction of Early-summer Northeast Asian Cut-off Lows by BCC-CSM2-HR and GloSea5. *Adv. Atmos. Sci.* 40, 2127-2134.

https://doi.org/10.1007/s00376-022-2197-9

- Nie, Yu, Yang Zhang, Jinqing Zuo, Mengling Wang, Jie Wu and Ying Liu, 2022: Dynamical processes controlling the evolution of early-summer cut-off lows in Northeast Asia. *Clim. Dyn*., 60, 1103-1119.

- Pinheiro, H., T. Ambrizzi, K. Hodges, M. Gan, K. Andrade, and J. Garcia, 2022: Are cut-off lows simulated better in CMIP6 compared to CMIP5? Climate Dyn., 59, 2117−2136, https://doi.org/10.1007/s00382-022-06200-9.

**Response to minor comment 1:** We thank the reviewer for pointing out this gap in our research. We agree that discussing the physical processes responsible for the COLs in South America is crucial for better understanding the origin of biases observed in the GEFS model. While our focus was on the forecast performance of COLs, we acknowledge that this discussion could help clarify the reason behind the intensity underestimation and the bias in position.

To address this, we have expanded the discussion section to incorporating insights from the literature, including the interaction between transient waves and the mean flow. Specifically, recent studies (Pinheiro et al., 2022; Nie et al., 2022, 2023) point to the weak representation of eddy-mean flow interactions as a key factor leading to forecast biases in COLs. These findings align with earlier work in Southern South America, which identified critical precursors for COL development. For instance, Godoy (2012) emphasizes the role of the jet stream's positioning and intensity, increased wave activity over the South Pacific, and the persistence of mid-latitude stationary wave patterns in shaping COL formation and evolution. These patterns are essential for understanding the forecast errors observed in GEFS predictions. While our current study focused on forecast performance, delving deeper into the simulation of jet streams and wave activity would enhance our understanding of these model biases.

Thus, to address the reviewer's comment, we have expanded the Discussion and Conclusion section to include the following:

*"In summary, the GEFS model performs well in predicting the onset of COLs up to three days in advance, but forecast skill diminishes with increasing lead time, with a notable westward bias and underestimation of intensity. This bias likely arises from the model's inadequate representation of eddy-mean flow interactions, as explored by Nie et al. (2022, 2023) and Pinheiro et al. (2022). Furthermore, in our study region, the positioning of the jet stream and*

*the enhancement of transient wave activity over the South Pacific, identified in previous work (Godoy, 2012), are key to understanding these biases. While a detailed investigation of the physical mechanisms underlying these forecast errors was beyond the scope of this study, we recognize that future work exploring the simulation of jet streams and Rossby wave activity could provide crucial insights. Preliminary research has already shown that specific Rossby wave patterns preceding COLs can be predicted up to a week in advance, although with reduced confidence beyond that period (Choquehuanca et al., 2023)."*

We believe that this expanded discussion addresses the reviewer's concern and enhances the manuscript by providing a more comprehensive understanding of the physical processes that influence COL predictability in Southern America.

**References of the response to minor comment 1:**

- Choquehuanca, B., Godoy, A., & Saurral, R. 2023: Evaluating cut-off lows forecast from the NCEP Global Ensemble Forecasting System (GEFS) in southern South America. Session 27: Hazards and Extreme events in The WCRP Open Science Conference ( Kigali, Rwanda)

- Godoy, A. A.: Procesos dinámicos asociados a las bajas segregadas en el sur de Sudamérica. Ph.D. thesis. Universidad de Buenos Aires. Facultad de Ciencias Exactas y Naturales, Argentina. https://hdl.handle.net/20.500.12110/tesis_n5602_Godoy , 2012.

2. **Minor comment 2:** Since the COLs often arise from the internal atmospheric variability similar to blocking high, I am curious about whether the ensemble spread of model is comparable to the variability of observed COLs.

   **Response to minor comment 2:** We thank the reviewer for this comment. However, in our approach we did not make use of ensemble members from GEFS, and focused our analysis on the ensemble mean. Therefore, we are not able to quantify the ensemble spread and to compare it against the observed variability of COLs.

   Still, it is worth noting that we found a significantly large correspondence between observed and forecast COL events in our dataset, which is indicative of the high skill of the GEFS model to predict these systems over southern South America.

3. **Minor comment 3:** In the abstract (Lines 22-23), the author stated that the depth and intensity of cold core can affect the thermodynamic instability pattern, precipitation and

horizontal temperature advection. I suggest the authors either add more related evidence to support these statements in the manuscript or adjust the statements in the abstract since the current manuscript is missing a detailed corresponding discussion.

**Response to minor comment 3:** Thank you for this comment. In the revised version, we have included additional references to provide more evidence supporting the influence of cold-core on thermodynamic instability, precipitation patterns and horizontal temperature advection. The discussion and the abstract in the manuscript have been revised and adjusted to reflect these changes.

The Discussion section is now expanded as follows:

*"It is well documented that the cold-core of COLs modulates the atmospheric instability response (Pinheiro et al., 2021; Hirota et al., 2016; Nieto et al., 2007; Porcu et al., 2007; Llasat et al., 2007; Palmen and Newton 1969). The dynamical ascent and atmospheric instabilities associated with the cold-core trigger and/or enhance precipitation events (Godoy et al., 2011; Nieto et al. 2007). "*

The abstract section is now rewritten as follows:

*"Cut-off Lows (COL) are mid-tropospheric cyclonic systems that frequently form over southern South America, where they can cause high-impact precipitation events. However, their prediction remains a challenging task, even in state-of-the-art numerical weather prediction systems. In this study, we assess the skill of the Global Ensemble Forecasting System (GEFS) in predicting COL formation and evolution over the South American region where the highest frequency and intensity of such events is observed. The target season is austral autumn (March to May), in which the frequency of these events maximizes. Results show that GEFS is skillful in predicting the onset of COLs up to 3 days ahead, even though forecasts initialized up to 7 days ahead may provide hints of COL formation. We also find that as the lead time increases, GEFS is affected by a systematic bias in which the forecast tracks lay to the west of their observed positions and intensity. Analysis of two case studies provide useful information on the mechanisms explaining the documented errors. These are mainly related to inaccuracies in forecasting the vertical structure of the two case studies, including their cold core and associated low-level circulation. These inaccuracies potentially affect thermodynamic instability patterns (thus shaping precipitation downstream) as well as the horizontal thermal advection which can act to reinforce or weaken the COLs. These results are expected to provide not only further insight into the physical processes at play in these*

*forecasts, but also useful tools to be used in operational forecasting of these high-impact weather events over southern South America."*

**References of the response to minor comment 3:**

- Godoy, A. A., Campetella M. C., Possia N. E.: Un caso de baja segregación en el sur de Sudamérica: Descripción del ciclo de vida y su relación con la precipitación. Revista Brasileira de Meteorologia. v.26, n.3, 491 - 502. doi: https://doi.org/10.1590/S0102-77862011000300014, 2011

- Hirota, N., Takayabu, Y. N., Kato, M., & Arakane, S. (2016). Roles of an atmospheric river and a cutoff low in the extreme precipitation event in Hiroshima on 19 August 2014. Monthly Weather Review, 144(3), 1145–1160. https://doi.org/10.1175/MWR-D-15-0299.1

- Llasat, M. C., Martín, F., & Barrera, A. (2007). From the concept of "Kaltlufttropfen" (cold air pool) to the cut-off low. The case of September 1971 in Spain as an example of their role in heavy rainfalls. Meteorology and Atmospheric Physics, 96, 43–60. https://doi.org/10.1007/s00703-006-0220-9

- Nieto, R., Gimeno, L., Añel, J., De la Torre, L., Gallego, D., Barriopedro, D., & Ribera, P. (2007). Analysis of the precipitation and cloudiness associated with COLs occurrence in the Iberian Peninsula. Meteorology and Atmospheric Physics, 96, 103–119. https://doi.org/10.1007/s00703-006-0223-6

- Palmén, E., & Newton, C. W. (1969). Atmospheric circulation systems: Their structure and physical interpretation. Academic Press.

- Pinheiro, H., Gan, M., & Hodges, K. (2021). Structure and evolution of intense austral cut-off lows. Quarterly Journal of the Royal Meteorological Society, 413. https://doi.org/10.1002/qj.3900

- Porcù, F., Carrassi, A., & Medaglia, C. (2007). A study on cut-off low vertical structure and precipitation in the Mediterranean region. Meteorology and Atmospheric Physics, 96, 121–140. https://doi.org/10.1007/s00703-006-0224-5

---

## Author Comment (AC2)

**Authors' response to reviewers' comments**

Manuscript Number: egusphere-2023-1996

Assessing the skill of high-impact weather forecasts in southern South America: a study on Cut-off Lows

The authors sincerely thank both reviewers for their detailed and constructive feedback, which significantly improved the manuscript. We have carefully considered all the comments and made substantial revisions to address the issues raised. These revisions have strengthened the analysis, particularly in relation to the predictability of Cut-Off Lows (COLs) and their impacts on Southern America during austral autumn. We believe the revised manuscript is now significantly clearer and more impactful.

In this document, the responses to the reviewers' comments are highlighted in red font, and the resulting corrected text is highlighted in green

**Reviewer 2**

This article analyses the predictability of cut-off lows during Autumn that frequent the south-west coast of South America. The authors analyse the forecast lead-time of cut-off low formation identified objectively in GEFS forecast data. Understanding the predictability of cut-off lows is important to understand, particularly in this part of the world, where they have high impact, as the authors suggest. There are however some gaps in how various analyses and processes are discussed in this work that need to be improved before publication.

Some general, specific and technical comments are below.

**General comments**

1. **General comment 1:** Methods non-specific (Section 2.3)

   The methodologies used within this study need to be clarified. I can understand that they are based on methodologies used and referenced in previous work, however at least some minor detail needs to be added to contextualise the results for the reader. This is especially true in Section 2.3. For example, "certain restrictions" - what are they?; "winds on the polar side of the labelled COL" - how far poleward?. These details are important for the reader to understand in order to understand the result presented

**Response to general comment 1:** We thank the reviewer for pointing out this gap in our research. We acknowledge that the initial version did not provide a comprehensive explanation in the methods section, especially in the tracking method (section 2.3), while we understand the significance of these details for the reader to interpret the results. Therefore in the revised version, we have changed the wording accordingly.

In particular

- We have clarified the "certain restrictions" mentioned in the initial manuscript regarding the detection of minimum geopotential heights at 300 hPa. Specifically, the algorithm looks whether the candidate point is at least 5 meters geopotential (mpg) lower than six surrounding grid points. If this criterion is not met, the algorithm evaluates fourteen surrounding points, requiring a 20 mpg difference to ensure closed circulation.

- The expression "winds on the polar side of the labeled COL" has been revised to specify the poleward extent considered. The wind shift is assessed at six grid points south of the candidate location.

- We have also provided additional context by referring to the conceptual framework from Nieto et al. (2005), which guides the methodology used for tracking COLs.

The revised Section 2.3 is now as follows:

*"**2.3 COL identification and tracking algorithm***

*The COLs dataset from GEFS and ERA5 is built following the approach outlined by Godoy (2012) and based on the conceptual framework of COL by Nieto et al. (2005). This conceptual model characterizes a COL as a closed cyclonic circulation isolated from the main westerly current and the presence of a cold core in mid-levels. We have chosen the 300 hPa level because it hosts both the largest frequencies and intensities of COLs within the Southern Hemisphere (e.g., Reboita et al., 2010; Pinheiro et al., 2021).*

*To detect COLs, the algorithm tracking uses the geopotential height and the zonal wind component at 300 hPa as well as the 300/850 hPa thickness, following a series of steps to classify potential grid points as COLs: Step 1) In order to impose the closed circulation, the algorithm looks for local minima in the 300 hPa geopotential height field. It selects a grid point that is at least 5 mpg lower than six of the eight surrounding grid points to ensure a higher geopotential height. If this condition is not met, the algorithm checks that fourteen out of the sixteen surrounding grid points have a higher or equal value within 20 mpg of the candidate grid point. Step 2) To ensure that the system is isolated from the westerly current, the algorithm looks for changes in wind direction at six grid points located south of the candidate grid point. Step 3) Finally, to confirm the presence of a cold core, the algorithm*

*employs the 850/300 hPa thickness as an indicator of temperature. It searches for a local minimum in thickness at the candidate point, following a procedure similar to the one used in the initial detection step. If a cold core is not found, the algorithm iterates through the eight surrounding grid points, accounting for possible displacements of the cold core relative to the geopotential minimum, as described in previous studies*

*For validation purposes, we performed a visual inspection of the ERA5 COLs outputs. This visual check confirmed that each event aligns with the conceptual model proposed by Nieto et al. (2005). Additionally, we stipulated that each COL should be identifiable for a minimum of two days in the reanalysis data. A total of 34 events met all the established criteria."*

**References to general comment 1:**

- Godoy, A. A.: Procesos dinámicos asociados a las bajas segregadas en el sur de Sudamérica. Ph.D. thesis. Universidad de Buenos Aires. Facultad de Ciencias Exactas y Naturales, Argentina. https://hdl.handle.net/20.500.12110/tesis_n5602_Godoy , 2012.

- Nieto, R., Gimeno, L., Laura de la Torre, Ribera, P., Gallego, D., García-Herrera, R., José Agustín García, Nuñez, M., Redaño, A., Jerónimo Lorente: Climatological Features of Cutoff Low Systems in the Northern Hemisphere. J. Climate, https://doi.org/https://doi.org/10.1175/JCLI3386.1 , 2005.

- Pinheiro, H., Gan, M., Hodges, K.: Structure and evolution of intense austral cut-off lows. Q. J. Roy. Meteor. Soc., https://doi.org/https://doi.org/10.1002/qj.3900, 2021.

- Reboita, M.S., Nieto, R., Gimeno, L., da Rocha, R.P., Ambrizzi, T., Garreaud, R., Krüger, L.F.: Climatological features of cutoff low systems in the Southern Hemisphere. J. Geophys. Res-Atmos., https://doi.org/https://doi.org/10.1029/2009JD013251, 2010.

2. **General Comment 2:** Verification metrics used

I have concerns that some of the verification metrics used may not be reflective of the processes they are trying to verify. This is particularly true of the CTE and ATE metrics. I am not sure that the ATE will consistently show an error in the speed of the track and CTE an error in the positional error of the track. An example situation of this is described in the specific comments below. Would showing speed and bearing not be more appropriate?

**Response to general comment 2:** Thank you for this important comment. We appreciate your concerns regarding ATE (along-track error) and CTE (cross-track error) metrics for

verifying the trajectory in COLs. The ATE and CTE metrics are commonly used for assessing the tracking of cyclones, however, we agree that these metrics have some limitations. For example, as you note in a hypothetical situation in which ATE is vastly underestimated, the forecasted speed of the cyclone is faster than the observed one (specific comment number 10). Therefore, these metrics do not always capture all aspects of the motion of COLs, but this does not necessarily invalidate our findings.

We agree that the use of other metrics such as speed and bearing metrics may give in some cases a better representation of the track errors. To illustrate this, we have computed the biases in velocity (i.e., the difference between forecasted and observed velocity of the COLs) and the bearing, defined as the angle formed by the forecasted and observed trajectories. These are shown in Fig. R1 and R2, respectively.

In the case of the velocity bias (Fig. R1), we observe positive values (indicative of overestimations of the actual speed in the forecasts) during initializations closest to the observed onset of the systems (Init 0). On the other hand, for initializations 2 or 3 days ahead of the actual segregation (Init 2 and Init 3, respectively) we obtain mostly negative values, suggesting that the model underestimates the actual speed of the systems. These results are well in line with what was obtained and discussed for the ATE (Fig. 8) regarding the mostly positive (negative) ATE values in days close to (far from) the dates of the observed COLs segregation.

Regarding the bearing (Fig. R2), results are overall less homogeneous across start dates. Overall, values are mostly positive during the first forecast days after segregation and become slightly negative at the longer lead times (days 5, 6 and 7 after segregation). A comparison of these results with those obtained for the CTE (Fig. 9) shows that the CTE values are also mostly positive at short lead times, regardless of the initialization day considered, which is in line with the bearing behavior.

In this regard, we conclude that results derived from the ATE and CTE metrics are solid and consistent, even more so after being compared to other independent metrics such as the velocity bias and the bearing.

[Figure]

Fig. R1 Velocity bias (i.e., the difference between forecasted and observed velocity of the COLs) as a function of the initialization days and for lead times 1 to 7 days after segregation. Units are km hr$^{-1}$.

[Figure]

Fig. R2 As in Fig. R1 but for the bearing, defined as the angle formed by the forecasted and observed trajectories. All angles are expressed in units of degrees.

3. **General comment 3:** Sample size

The sample of COLs used in this work seems incredibly small given the number of COLs that this region receives - this represents less than 1 COL per year. I wonder therefore if the region or criteria is too strong in this study. Relaxing it and getting more events would help significantly in making sure the results are more robust. I was also unclear exactly how COLs in the study region are selected - is it only COLs that begin (as Stage 2) in the region that are selected? Or are COLs that move through the region as Stage 2 also counted?

**Response to general comment 3:** The main reason our sample is smaller than expected is that we applied several criteria for the selection of COLs and region study. These criteria on the one hand lead to a substantial reduction in the number of systems, while on the other hand ensure the detection of relevant COLs. Let us clarify these points:

- Specific region: We focused on a specific region located west of the coasts of central Chile, in southwestern South America. This region has been previously identified as the one with the largest frequency of COLs (Barnes et al., 2021; Pinheiro et al., 2017; Campetella and Possia, 2007). It is also the region where COLs forming there have a more significant impact on populated areas of southern South America (Godoy et al., 2011). We could have increased the size of our sample, but at the expense of including systems with very little or no affectation to land areas (which was one of the main objectives of our study)

- Specific season: We only considered systems developing during the austral autumn (from March to May). That is the season in which the highest number of systems develop, and also when the more significant impacts on land areas are detected.

- Specific duration: We imposed a restriction on the number of days a COL is detected. By this criterion, we discarded COLs lasting less than 2 days.

It is also worth noting that systems crossing the box of COLs formation but not forming within that area were also not included in our analysis. This is because we are also aiming at identifying the key physical mechanisms responsible for the formation and onset of COLs.

**References to general comment 3:**

- Barnes, M. A., Ndarana, T., & Landman, W. A. (2021). Cut-off lows in the Southern Hemisphere and their extension to the surface. Climate Dynamics, 56(11-12), 3709–3732. https://doi.org/10.1007/s00382-021-05662-7
- Campetella, C. M., & Possia, N. E. (2007). Upper-level cut-off lows in southern South America. Meteorology and Atmospheric Physics, 96(3-4), 181–191. https://doi.org/10.1007/s00703-006-0227-2

4. **General Comment 4:** Some conclusions reached seem to lack evidence

   The authors make several conclusions related to the precipitation patterns, depth or vertical coupling and thermodynamics of the COLs related to errors in the forecast of these systems. Many of these are expressed as important results in both the conclusion and the abstract, however, in my view, these largely have very little evidence to back them up. These are largely based on two individual case studies of COL simulations by analysis of the ensemble mean. Can holistic conclusions be drawn for just these two cases? Statements such as "predictive skill of COLs had a significant impact on the precipitation forecasts downstream" or "underestimating the strength of the cold core of COLs can significantly alter the vertical coupling of COLs" need to be robustly backed up by the results. One way to make this more robus would be analyse each of the members and leverage the ensemble spread to understand the processes associated with small and large positional errors.

   **Response to general comment 4:** Thank you for your feedback. We would like to clarify your concerns about the robustness of our conclusions in section 3.3, related to the precipitation patterns, the vertical coupling and the thermodynamics of COLs.

   We acknowledge that our research (Section 3.3) is based on only two case studies, which is limited to drawing holistic conclusions. However, it is important to note that our goal is not to make generalizations about COLs. Instead, we seek to provide an initial framework for understanding how the specific physical characteristics of COLs west of the Andes mountains may influence weather forecasting, particularly in terms of precipitation patterns.

   Both cases were chosen because they exhibited significant differences in their three-dimensional structure before and during their life cycle, indicating that they may have been influenced by different dynamic forcings (Pinheiro et al., 2021). The first case is characterized by its small horizontal extent and the absence of well-defined large-scale features, suggesting weaker dynamic forcing in its formation. Conversely, the second case displays marked large-scale features with a well-defined structure. These cases were selected due to their representative characteristics and complexities that encompass a wide range of behaviors typical of COLs in the studied region. We are aware of the need to evaluate a larger number of cases to reinforce the generalizability of our results.

   Regarding affirmations such as " predictive skill of COLs had a significant impact on the precipitation forecasts downstream" or "underestimating the strength of the cold core of COLs can significantly alter the vertical coupling of COLs" we have improved the wording as follows:

*"Although this study is based on only two case studies, our analysis indicates that the predictive skill of COLs, particularly regarding their formation location, intensity, and trajectory, can influence precipitation forecasts downstream."*

*"In our case studies, underestimating the strength of the cold core of COLs pottential affect thermodynamic instability patterns, potentially influencing vertical motion and precipitation formation downstream."*

**References to general comment 4:**

- Pinheiro, H., Gan, M., & Hodges, K. (2021). Structure and evolution of intense austral cut-off lows. Quarterly Journal of the Royal Meteorological Society. https://doi.org/10.1002/qj.3900

**Specific comments**

1. L70: The precipitation patterns studied seem very local and not "downstream". In my view therefore, this posed question is not answered as stated.

Thanks for this observation. Note that the location of precipitation relative to the COLs depends on the dynamical mechanisms that cause the precipitation. Palmen and Newton (1969) show that the general distribution of precipitation patterns associated with COLs has a remarkable asymmetric structure, with precipitation on the east side of the cyclone (Fig. 10.6 of Palmen and Newton, 1969). In our research, we considered the area of influence of the COLs, approximately 7° degrees (about 700 km radius) from the geopotential height minimum at 300 hPa, which corresponds with the typical diameter of COL systems (Kentarchos and Davies, 1998). Therefore, we believe that this approximation accurately addresses the question of "downstream" influence as in the original statement in L70. While we may not have sufficiently clarified this in the first version, we have emphasized this focus in our revised manuscript (please see section 3.3).

The text in the revised version is now as follows:

*"3.3 Case studies*

*In this subsection, we focus on two COLs that exhibited very different levels of prediction performance during their onset stage (Fig. 4a). The first case study, from March-April 2013, is characterized by small DPE values, below the first quartile in Fig. 4a, indicative of a forecast with high accuracy in the GEFS dataset. In contrast, the second case study, from March 2019, was associated with remarkably larger DPE values, with errors ranging between the median and the third quartile. This represents a scenario in which the prediction*

*has a suboptimal performance. It is important to note that the selection of the case studies was based also on the impact the model errors had on the associated precipitation downstream. For the analysis of precipitation, we considered as the area of influence of the COLs approximately 7 degrees (about ~700 km radius) from the geopotential height minimum at 300 hPa. Before exploring the associated errors in the GEFS dataset, we provide a brief description of the synoptic environment around each COL during its segregation stage."*

Reference specific comments 1

- Palmén, E., & Newton, C. W. (1969). *Atmospheric circulation systems: Their structure and physical interpretation.* Academic Press.
- Kentarchos, A. S., & Davies, T. D. (1998). A climatology of cut-off lows at 200 hPa in the Northern Hemisphere, 1990–1994. *International Journal of Climatology, 18*(4), 379–390.https://doi.org/10.1002/(SICI)1097-0088(19980330)18:4<379::AID-JOC257>3.0.CO;2-F

2. L88: Previous authors (e.g. Reboita et al. (2010)) have found large differences in frequencies of COLs for different reanalyses, with ERA reanalyses producing more COLs than NCEP reanalyses. Since the authors use GEFS, would the use of NCEP reanalysis not be a fairer comparison?

To respond to this comment, we compared our COLs dataset using ERA5 with the COLs dataset of Godoy (2012) using NCEP-NCAR reanalysis. This comparison focused on the autumn seasons between 1999 and 2010, the period studied by Godoy (2012). We found a good correspondence in number and in location between both COL datasets. While our COL dataset detected 12 COLs, Godoy (2012) found 11 COLs, which can be related to the higher resolution of modern reanalysis as ERA5 than with the older NCEP-NCAR. Recently, Pinheiro et al. (2019) showed that the spatial differences observed between the newest reanalyses are mainly due to differences in the track lengths, which is larger in ERA-Interim than in NCEP-CFSR.

Reference of specific comment 2:
- Godoy, A. A.: Procesos dinámicos asociados a las bajas segregadas en el sur de Sudamérica. Ph.D. thesis. Universidad de Buenos Aires. Facultad de Ciencias Exactas y Naturales, Argentina. https://hdl.handle.net/20.500.12110/tesis_n5602_Godoy , 2012.

- Pinheiro,H.R., Hodges, K.I. & Gan, M.A. An intercomparison of subtropical cut-off lows in the Southern Hemisphere using recent reanalyses: ERA-Interim, NCEP-CFRS, MERRA-2, JRA-55, and JRA-25. Clim Dyn 54, 777–792 (2020). https://doi.org/10.1007/s00382-019-05089-1

3. L90-93: Does the orography affect your choice of using 850hPa in your thickness criteria? Presumably, a large proportion of the Andes is generally above 850hPa.

In this case, the orography does not have a significant impact. As mentioned in our paper, the majority of the COL systems have formed to the west of the Andes mountains and have not been affected by any orographic effect during their formation. It's important to note that the Andes are a relatively narrow range with a typical width of less than 200 km, significantly limiting any possible orographic impact on the COL systems (Garreaud 2009).

References to specific comment 3
- Garreaud, R. D.: The Andes climate and weather, Adv. Geosci., 22, 3–11, https://doi.org/10.5194/adgeo-22-3-2009, 2009.

4. L94: I assume the "accumulated precipitation" is ERA5 data. General I have found that these totals are generally error prone. Is ERA5 precipitation reliable?

We have revised the recent literature and found that Lavers et al. (2022) found that ERA5 precipitation agrees very well with pluviometers data in west southwest South America, especially during autumn. Therefore, we considered it a trustful dataset against which to compare the predicted rainfall amounts.

Reference:
- Lavers, D.A., Simmons, A., Vamborg, F. & Rodwell, M.J.(2022) An evaluation of ERA5 precipitation for climate monitoring. *Quarterly Journal of the Royal Meteorological Society*, 148(748) 3124–3137. Available from: https://doi.org/10.1002/qj.4351

5. L108: What are these restrictions?

The restrictions have been clarified in the revised manuscript. See Response Comment 1, for further details.

**6.** L109: I assume you look at a minimum in thickness? Provide details on this procedure

Yes, we use the minimum thickness to identify the cold core. The sentence has been revised for clarity and now reads in the revised manuscript as follows:

*"Finally, to confirm the presence of a cold core, the algorithm employs the 850/300 hPa thickness as an indicator of temperature. It searches for a local minimum in thickness at the candidate point, following a procedure similar to the one used in the initial detection step. If a cold core is not found, the algorithm iterates through the eight surrounding grid points, accounting for possible displacements of the cold core relative to the geopotential minimum, as described in previous studies."*

**7.** L111: How far poleward?

This has been clarified in the revised manuscript. See  Response General Comment 1,  for further details.

**8.** L113: What are these specific criteria?

The specific criteria refer to the spatial and temporal thresholds used to match the GEFS COL dataset with the ERA5 COL dataset. To clarify this, we have revised the corresponding paragraph to provide more detail on the matching process.

The criteria are as follows:

- **Spatial criterion**: This refers to the maximum allowable distance between the forecasted GEFS COL and the observed ERA5 COL trajectories. Specifically, a GEFS COL was considered to correspond to an ERA5 COL if the initial points of both trajectories were within 800 kilometers of each other. This distance was chosen based on the typical diameter of COL systems, which ranges between 600 and 1200 kilometers (Kentarchos and Davies, 1998). We focused on the first point of the forecasted trajectory, rather than the entire trajectory, due to the likelihood of increasing divergence as forecast lead times extend.

- **Temporal criterion**: This refers to the number of days during which the GEFS and ERA5 COL trajectories overlapped. A match was considered valid if at least one point along the forecasted system's life cycle coincided in time with the ERA5 trajectory, ensuring that the systems were contemporaneous for at least one day.

To enhance clarity, we have rewritten the paragraph in the manuscript as follows:

*"Following the identification of the COLs, we validated the GEFS COL dataset by comparing it with the ERA5 COL dataset. A GEFS COL was considered to correspond to the same system as in the ERA5 COL dataset if their respective trajectories satisfied predefined spatial*

*and temporal criteria. The forecasted COL trajectories that met these criteria were used to generate diagnostics, quantifying errors in predicted positions, intensities, and other properties of the COLs. The spatial criterion required that the distance between the forecasted and reanalysis trajectories did not exceed 800 kilometers—this threshold was chosen based on the typical diameter of COL systems, which ranges between 600 and 1200 kilometers (Kentarchos and Davies, 1998). Notably, our spatial criterion primarily focuses on the initial segment of the forecast trajectories rather than the entire track, consistent with the methodology of Froude et al. (2007). This approach is justified by the expectation that forecast accuracy is generally higher at the start of the trajectory, where GEFS trajectories are likely to be more closely aligned with their ERA5 counterparts. Regarding the temporal criterion, a match was considered valid if at least one point along the system's life cycle coincided in time (i.e., within a 24-hour period)."*

9. L117-120: Not sure I understand this point. Consider clarifying.

   We have clarified this point in the revised manuscript. Specifically, we emphasize that the spatial criteria in our analysis focuses on the initial portion of the forecast trajectories, rather than the entire track, following the approach of Froude et al. (2007). This criterion is applied only at the starting point, where forecast performance is generally expected to be more accurate and therefore GEFS trajectories may begin very close to the corresponding trajectories in the ERA5 dataset. See Response specific comment 8, for further details.

   Reference specific comments 1
   - Froude, L. S. R., Bengtsson, L., & Hodges, K. I.: The Predictability of Extratropical Storm Tracks and the Sensitivity of Their Prediction to the Observing System, Mon. Weather Rev., 135(2), 315-333. doi: https://doi.org/10.1175/MWR3274.1, 2007

10. L128-132: How robust is the use of ATE to represent speed and CTE to represent spatial error? I can envisage situations in which the along track error vastly under-estimates the speed of propagation (see my poor recreation of your Fig 2. below). In this case, the speed error is presumably under-estimated if one looks at the ATE, however in reality the speed is actually faster than in reality. Would using speed/distance travelled (to determine speed errors) and bearing (to represent bias in left/right not be better?

[Figure]

This issue was clarified in the second general comment above. See Response of general Comment 2, for further details.

11. L139: How is intensity defined in this study?

In our study, the intensity of COLs is quantified by calculating the Laplacian of the geopotential height field. Specifically, we define intensity as the peak value of the Laplacian that spatially coincides with the center of the COL. We have revised line L139 to the following:

*"The intensity of COLs is defined by the maximum value of the Laplacian of the geopotential height field, where this maximum corresponds to the location of the center of COLs."*

12. L139-141: "7 days before" - Do you refer to forecast lead-times here? Clarify this statement.

To enhance clarity, we have revised the text as follows:

*"We present results for forecasts initialized up to seven days prior to the observed onset of COL events, as the preliminary analysis indicated that no COLs were forecasted beyond this lead time."*

13. L149: 34 COLs - seems a small number (~1 COL per season). Would increasing your domain size give you more COLs and increase the robustness of your results? This number is too small to get a robust result in my view

This one point has been clarified in the revised manuscript. See Response to general comment 3, for further details.

**14.** Figs 6-9: I strongly suggest that these figures are combined into a single plot to help the reader compare these different metrics. Either stack each metric on an init axis,or stack inits on a metric axis.

We agree. It has been changed as suggested.

**15.** L178-179: Do all your tracks last at least 3 days in order to study the track evolution after 3 days? Most climatologies (some of which are cited here) shows the majority of COLs last for only 1-2 days. Please clarify.

Thank you for this observation. You are right, most global climatologies show that the COLs typically persist for only 1-2 days. However, the focus of our study is the west coast of southern South America, a region where COLs tend to exhibit greater intensity compared to other areas (Pinheiro et al., 2021; Godoy, 2012 ; Barnes et al., 2021) and therefore these COLs tend to last longer (Barnes et al., 2021; Godoy, 2012). o clarify this point, we have specifically calculated the lifespan of the COLs identified in our analysis, defining their duration from the onset of segregation to the dissipation phase. As a result, we find that out of the total COLs identified in this study, 48.57% (17 COLs) have a duration of 3-4 days, 31.43% (11 BS) have a duration of 5-8 days, and 20% (7 BS) have a duration of 2 days (Fig. R4).

[Figure]

Fig. R3: Number of COLs duration.

To clarify this point for readers we have added a brief mention of this analysis in the revised paper.

*"Given that a preliminary study shows that a large portion of COLs in the study region have lifespans of 3–4 days or more, with nearly 80% lasting beyond 3 days (not shown), we have focused our analysis on forecast lead times of up to 3 days following the initial detection of these COLs in the ERA5 reanalysis."*

References to specific comment 15:

- Barnes, M. A., Ndarana, T., & Landman, W. A. (2021). Cut-off lows in the Southern Hemisphere and their extension to the surface. Climate Dynamics, 56(11-12), 3709–3732. https://doi.org/10.1007/s00382-021-05662-7
- Godoy, A. A.: Procesos dinámicos asociados a las bajas segregadas en el sur de Sudamérica. Ph.D. thesis. Universidad de Buenos Aires. Facultad de Ciencias Exactas y Naturales, Argentina. https://hdl.handle.net/20.500.12110/tesis_n5602_Godoy , 2012.

- Pinheiro, H., Gan, M., Hodges, K.: Structure and evolution of intense austral cut-off lows. Q. J. Roy. Meteor. Soc., https://doi.org/https://doi.org/10.1002/qj.3900, 2021.

16. Section 3.2: Forecast COLs are generally weaker than observed. How does this impact their predictability? Do they generally terminate more quickly than observed as they are weaker? This is an interesting observation. The intensity of COLs is a crucial factor that impacts their structure; most intense COLs show a deep vertical structure, while less intense COLs show a shallow vertical structure (Pinheiro et al. 2021). Therefore our result (the systematic weakness of COLs in GEFS) is likely linked with their predictability. In order to answer this question we analyzed the distribution of intensity values of COLs (the maximum value of intensity in the lifespan of COLs) and their duration, respectively (Fig. R4). Fig. R4 shows that as lead time increases, GEFS tends to predict shorter systems. This is evident in the evolution of GEFS median in comparison with ERA5 median. A similar behavior is observed by intensity distribution, as lead time increases, GEFS predicts weaker systems than ERA5. In summary, a less accurate prediction of COL intensity impacts the accuracy of forecasting COL lifespan.

References of specific comment 16:

- Pinheiro, H., Gan, M., Hodges, K.: Structure and evolution of intense austral cut-off lows. Q. J. Roy. Meteor. Soc., https://doi.org/https://doi.org/10.1002/qj.3900, 2021.

[Figure]

Fig. R4: On the left: the distribution of duration of COLs from ERA5 reanalysis (blue bars) and GEFS reforecast (red bars). On the right: the distribution intensity of COLs from ERA5 reanalysis (green bars) and GEFS reforecast (yellow bars)

**17.** L192: "except for init 2" - init 0 also fairly symmetric

We agree. It has been corrected accordingly as follow:

*"The ATE distribution exhibits a negative bias towards the later stages of the forecast trajectories, except for init 2 and init 0 (Fig. 8c) which show slightly positive values."*

**18.** L217-220 and 240-243: This synoptic description is unnecessary. This circulation is there by definition of your COL identification technique.

We think that it is useful to keep these paragraphs, as they help to provide a description of the physical processes associated with the COLs, particularly their mechanisms of formation and sustainment. While the circulation is inherent to our COL identification method, the detailed description offers valuable context for readers unfamiliar with these dynamics. Furthermore, the inclusion of these synoptic features addresses a concern raised by the other anonymous reviewer, who specifically recommended their discussion and inclusion in the manuscript.

**19.** L226-228: It is unclear from your analysis how the vertical coupling was affected by the COLs incorrect forecast position. Be specific and add detail to this analysis.

We have revised and corrected the text to address this point based on your suggestion. In particular, we focused on how the incorrect position of the low-level cyclonic circulation affects the vertical coupling. We have also discussed how this incorrect forecast weakens the vertical motion and the moisture convergence. The updated version of the text is now as follows:

Correction:

*"Regarding background circulation, the GEFS model accurately predicted the strength and extent of upper high winds associated with the COL (Fig. 10d,g). However, it underestimated the strength of the cold core in middle levels. Furthermore, the cyclonic circulation at lower levels is displaced to the north relative to the observation (Fig. 10c), which explains why the COL and lower-level cyclones are out of phase. This results in a different vertical structure in the forecasts with regard to the observations, which is consistent with the underestimation of the COLs intensity in the model. As discussed by Pinheiro et al. (2021), the intensity of the COL directly affects the COL vertical structure. In this case, the incorrect forecast position of the lower cyclone likely weakened the upward vertical motion and low-level moisture convergence, both of which are key factors for precipitation development. This implies a weaker vertical coupling in the forecast, resulting from the discrepancy in the intensity of the COL."*

**20.** Figures 10-11: Both Sections 3.3.1 and 3.3.2 make reference to low-level circulation patterns, however these are not shown.

The caption of these figures has been corrected.

**21.** L231-233: I disagree from Figure 10, that one could say that GEFS had "difficulties" in predicting the split jet and "failed" to capture the cold-core. Both these features are present in Figure 10j,k,l

We have revised the manuscript based on the reviewer's comments to better clarify our findings:

*"In terms of background circulation, the GEFS model accurately predicted the strength and extent of the upper-level high winds associated with the COL (jet-split structure) and the upstream ridge of the COL for init 1, init 3 and init 5 (Fig. 10d, g, j). Particularly, during init 5 (Fig. 10j) it predicts better the intensity of the jet streak on the polar side of COL than the jet on the equatorial side. In the mid-levels, the model successfully captured the cold core during init 1 and init 3, although with slightly less strength compared to ERA5 reanalysis. However, it failed to capture the cold core during init 5."*

**22.** L235: Do you feel that this is a robust statement? In general GEFS seems to have done a decent job. Your longest lead-time produced a weaker COL in the mean which affected the forecast of higher precipitation. However I feel that the statement that GEFS (which is relatively course resolution here) may not perform well for COL precipitation forecasts is very harsh given the data provided and the severely limited cases looked at.

We have revised the manuscript in response to the reviewer's comments to improve clarity and provide a more balanced interpretation of GEFS performance. The revised statement now reads as follows:

*"Regarding precipitation forecasts, GEFS performs well in predicting the location of precipitation associated with COL (with a slightly southeast bias), but it underestimates the amount of precipitation, especially during init 3 and init 5, with underestimations around 20 mm/day (Fig. 10h,k)."*

**23.** L244-246: Is this frontal zone not linked at all to the COL? Ie. Is this front not connected to the surface low (if there is one) associated with the COL or is it linked to a completely separate mid-latitude cyclone? Be specific.

The COL analyzed in this part of the manuscript did not have a surface front. The decaying frontal system we are referring to is in fact linked to a surface cyclone but well away from our

study area, over the South Atlantic Ocean, which is a typical area of frontolysis over southern South America. We have clarified this in the text as follows:

*"In the precipitation field, two distinct maxima were identified: one located northeast of the analysis domain, associated with a decaying frontal zone in that area, which is linked to a surface cyclone positioned over the South Atlantic Ocean (not shown), and another maximum over western Argentina, directly related to the ascent zone east of the COL. The frontal system mentioned here is separate from the COL and its associated dynamics."*

24. L251: "the circulation at low and mid-levels" - how do we know this? This implies that the low and mid-level circulation are completely independent of the upper-levels, which they are not of course. Clarify and add detail to explain your analysis

We have revised the manuscript to clarify the relationship between circulation at different levels. The revised text is as follows:

*"With respect to the upper-level winds associated with the COL, the GEFS demonstrated strong skill in forecasting both the intensity and spatial positioning of these winds, particularly in relation to jet streaks on the polar flank of the COL. However, the model exhibited notable challenges in accurately representing the cold-core structure at mid-levels, with a complete absence of this feature in the init. 5. At lower levels, the representation of the closed cyclone at 850 hPa was similarly problematic, with the system consistently displaced northward and exhibiting weaker intensity than observations, especially in the init. 3 and 5"*

25. L257-259: Similarly to case 1, I think the rainfall patterns presented look pretty good, especially considering that the model is 1x1 degree. What is the effect of the resolution on your analysis?

Thanks for this comment. In effect, the resolution of the GEFS model certainly plays a role in the representation of precipitation. This could be limiting its ability to capture fine-scale features in complex terrain, especially at the lee side of the Andes mountains. Previous studies have shown that even higher-resolution models have difficulties in accurately representing precipitation in these areas (Yañez-Morroni et al., 2018). These limitations are highlighted in the revised manuscript to provide context for observed discrepancies.

*"In terms of precipitation, GEFS underestimated rainfall amounts in all initializations and cannot represent the precipitation at the lee side of Andes mountain (Fig. 11e,h,k), displacing*

*the predicted precipitation northeast of the observed location, particularly over central and northeastern Argentina. Howevere, while the GEFS model generally underestimated rainfall amounts across all initializations, it is important to note that this behavior is expected given the model's relatively coarse resolution (1x1 degree), specially at lee side of Andes mountain where the complex features of COLs usually difficult the simulation of precipitation even to hight resolution models like WRF (Yañe-Morroni et al., 2018). "*

Reference of specific comment 25:

- Yáñez-Morroni, G., Gironás, J., Caneo, M., Delgado, R., Garreaud, R.: Using the Weather Research and Forecasting (WRF) Model for Precipitation Forecasting in an Andean Region with Complex Topography. Atmosphere. https://doi.org/10.3390/atmos9080304, 2018

26. L262: The final row of Fig 11 appears not to be cut-off, but called a COL here. Would this have been detected as a COL in your analysis?
   We agree. It has been changed as suggested.

Technical comments

- L32: "suppose" -> "pose"?
  It has been changed as suggested.

- L82-83: is data from the second week used here? It seems that only lead-times up to 7 days are investigated. If not, remove this detail.
  We agree this has been removed

- L137: "position is biased fast (slow)" - consider rewording
  Agree, this has been changed and rewritten for clarity.

- L168: "Conversely,..." - this sentence is not converse to the previous statements. Both indicate large spread, the one is simply larger than the other.
  Agree, It has been changed

- L172: "direction" -> "bias"?
  Agree, it has been changed

- Figure 10-11: Maybe "Obs" is not a representative label, but rather "Reanalysis".

The figure label has been changed as suggested.

- L176 and elsewhere: You use the terminology "segregation", "onset" and "start" throughout the article. It remains unclear whether these are different or mean the same thing. If there are the same thing, stick to one terminology. If not, define them explicitly and clarify.

  These terms are used interchangeably and refer to the same concept. This is clarified in lines 141-142.

- L182: "ERA" -> "ERA5"?

  It has been changed as suggested

- L187-202: "increases" - do you mean "decreases"? The value gets less. The magnitude of the error increases. Be specific.

  Agree, this has been changed and rewritten for clarity.

- L221: "record-breaking rainfall": 25-50mm seems low for a rainfall record. If this is the record, then add a reference to backup this statement.

  Agree, this has been changed and rewritten for clarity.

- L243: "northeast of the country" - which country?

  It has been changed for clarity

- L238: "the previous COL" -> "Case 1"

  Agree, this has been changed

- L243-244: "Figure 11c" - this is thickness and geopotential height at 300hPa and not low-level circulation as specified.

  We have revised the figure and corrected the corresponding title to accurately reflect the data presented.

- L282: "have a prominent" -> "tend to have a" - the bias is not that prominent is it

  Agree, this has been changed

---

## Author Response (AR3)

**Response to Co-editor's comment**

Manuscript Number: egusphere-2023-1996 Assessing the skill of high-impact weather forecasts in southern South America: a study on Cut-off Lows

**Co-editor comment**

Public justification (visible to the public if the article is accepted and published):

Thank you for your detailed revision and reply to the reviewers' comments, through which the quality of the manuscript has been substantially improved. Based on the second round of reviews and my own evaluation, I think the manuscript is acceptable for publication after some minor revision. Here are my suggestions on paper revision for consideration. (The line numbers in my following suggestions are based on the changes tracked version of the manuscript.)

1. I appreciate the authors' effort in adding more discussion on the dynamics underlying the prediction skills. However, I found some arguments in the discussion are with a strong tone but lacking sufficient support. I suggest the authors either add more evidence to support the arguments or revise some of their arguments in the discussion. For example: in the conclusion part (Lines 378-383), the authors argue that "In our case studies, the strength of the COLs cold core affects the thermodynamic instability patterns, potentially influencing vertical motion and precipitation formation downstream." I agree with the authors that the two case studies all suggest the deficiency of models in simulating the vertical structure especially the low-level circulations of the COLs, but you did not do further investigations on how this deficiency affects the thermodynamic instability as well as the vertical motion, though based on previous studies this is likely to happen. The authors did not discuss this point in the text when introducing their results of case studies either. I suggest the authors either add more analysis on the thermodynamic instability and vertical motion in the two cases or revise the arguments here.

2. Figures 7 and 8: The right column of the figures show the 850-300 hPa layer thickness and the 850 hPa geopotential height of reanalysis and GEFS prediction. However, the labels of the contours indicating geopotential height in these subfigures are very hard to read. The authors may need to change the contour interval, weight or color, the contour thickness or so on to make it easier to read. BTW, Line 578, it should be "850-300 hPa", instead of "850/3000 hPa". And I suggest the authors to indicate contour intervals in the figure caption.

3. The third point of your conclusion (Lines 373-392) is mainly based on the two case studies. It needs to test in the future that whether the conclusions made in the two case studies still hold in other cases. This point needs to be emphasized in the text. Meanwhile, the expression in Line 373 "Although this study is based on only two case studies, our analysis suggests that…" better be changed to "Our two case studies suggest that…"

Additional private note (visible to authors and reviewers only):

Thank you for your patience in the review process!

**Author's Response**

The authors sincerely appreciate the feedback provided by the co-editor. We value your acknowledgment of our efforts to enhance the manuscript and welcome your constructive suggestions for further refinement. Below, we address each of your comments in detail:

1. **Tone of arguments in Lines 378-383**

   Thank you for your feedback about the tone of our arguments in Lines 378-383. We understand the importance of being more careful with our conclusions. While prior studies suggest these connections, our analysis does not directly explore these mechanisms. To address this, we have.

   - Revised the phrasing in Lines 378-383 to highlight that these links are potential rather than definitive.

   - Added a statement acknowledging the need for additional research to better understand and confirm these mechanisms in the context of COLs in South America.

   *Changes made in the manuscript:*

   - Page 11, Lines 378-383: We have improved the wording as follows:

   "In particular, the errors in the location and depth of the COLs appear to be linked to the mechanism sustaining these systems. In our case studies, the strength of the COLs cold core could affect the thermodynamic instability patterns, potentially influencing vertical motion and precipitation formation downstream, even though further research would be needed to assess the actual role of the mechanisms at play. This is also further supported by the well-documented relationship between COLs cold-core and atmospheric instability response (Pinheiro et al., 2021; Hirota et al., 2016; Nieto et al., 2007; Porcu et al., 2007; Llasat et al., 2007; Palmen and Newton 1969), through which the dynamical ascent and atmospheric instability associated with the cold-core trigger and/or enhance precipitation events (Godoy et al., 2011; Nieto et al. 2007)."

2. **Readability of Figures 7 and 8**

   Thank you for highlighting the readability issues with Figures 7 and 8. We have made the following adjustments:

   - Increased the contour thickness and adjusted the intervals to enhance clarity.

   - Specified the contour intervals in the figure captions as suggested.

   - Corrected the typographical error on Line 578, changing "850/3000 hPa" to "850-300 hPa."

   These modifications enhance the interpretability of the figures and ensure consistency with the overall quality of the manuscript.

**3. Emphasis on the limitations of two case studies**

We agree with your suggestion to emphasize the limitations of basing conclusions on two case studies. To address this, we have:

- Rephrased the statement in Line 373 to reflect the scope of the study more accurately.

- Added a note in the conclusions section to highlight the need for further studies to test the generality of our findings.

*Changes made in the manuscript:*

- Page 11, Line 373: Revised the sentence:

  "Although this study is based on only two case studies, our analysis suggests that…"

  to:

  "Our two case studies suggest that…"

- Page 11, Line 392: Added the statement: "It should be noted however that these conclusions are driven by two case studies, and more research dealing with the processes associated with COL formation are needed. "

We believe these revisions address your comments comprehensively and improve the clarity of the manuscript. Thank you for your guidance and support throughout the review process.

Sincerely,

Belén Choquehuanca

On behalf of all co-authors